# Inhibition mechanism of SARS-CoV-2 main protease by ebselen and its derivatives

Kangsa Amporndanai[1], Xiaoli Meng [2], Weijuan Shang[3], Zhenmig Jin [4], Michael Rogers[5], Yao Zhao[4], Zihe Rao [4], Zhi-Jie Liu [6], Haitao Yang [4✉], Leike Zhang [3✉], Paul M. O'Neill [5✉] & S. Samar Hasnain [1✉]

The SARS-CoV-2 pandemic has triggered global efforts to develop therapeutics. The main protease of SARS-CoV-2 (Mpro), critical for viral replication, is a key target for therapeutic development. An organoselenium drug called ebselen has been demonstrated to have potent Mpro inhibition and antiviral activity. We have examined the binding modes of ebselen and its derivative in Mpro via high resolution co-crystallography and investigated their chemical reactivity via mass spectrometry. Stronger Mpro inhibition than ebselen and potent ability to rescue infected cells were observed for a number of derivatives. A free selenium atom bound with cysteine of catalytic dyad has been revealed in crystallographic structures of Mpro with ebselen and MR6-31-2 suggesting hydrolysis of the enzyme bound organoselenium covalent adduct and formation of a phenolic by-product, confirmed by mass spectrometry. The target engagement with selenation mechanism of inhibition suggests wider therapeutic applications of these compounds against SARS-CoV-2 and other zoonotic *beta*-corona viruses.

[1] Molecular Biophysics Group, Department of Biochemistry and System Biology, Institute of System, Molecular and Integrative Biology, Faculty of Health and Life Sciences, University of Liverpool, Liverpool L69 7ZB, UK. [2] Department of Molecular and Clinical Pharmacology, Institute of Translational Medicine, Faculty of Health and Life Sciences, University of Liverpool, Liverpool L69 3BX, UK. [3] State Key Laboratory of Virology, Wuhan Institute of Virology, Chinese Academy of Sciences, Wuhan 430071 Hubei, China. [4] Shanghai Institute for Advanced Immunochemical Studies and School of Life Science and Technology, ShanghaiTech University, Shanghai 201210, China. [5] Department of Chemistry, Faculty of Science and Engineering, University of Liverpool, Liverpool L69 7ZD, UK. [6] iHuman Institute and School of Life Science and Technology, ShanghaiTech University, Shanghai 201210, China. ✉email: yanght@shanghaitech.edu.cn; zhangleike@wh.iov.cn; P.M.Oneill01@liverpool.ac.uk; S.S.Hasnain@liverpool.ac.uk

The recent emergence of severe acute respiratory syndrome coronavirus 2 (SARS-CoV-2) has resulted in a global pandemic of coronavirus disease 2019 (COVID-19) with confirmed infection cases of over 140 million and 3 million fatalities as of April 2021. SARS-CoV-2 is the most devastating zoonotic coronavirus to infect humans following SARS-CoV-1 and MERS-CoV (Middle East respiratory syndrome) which emerged in 2002 and 2012, respectively[1]. Similar to the other coronaviruses, SARS-CoV-2 primarily infects the respiratory system and develops critical pneumonia that highly necessitates ventilatory support and intensive care, particularly in elderly and immunocompromised individuals[2]. Whilst there have been tremendous strides forward in the development of vaccines, the current roll-out is supply and time-limited. Several vaccines have been developed and approved for mass immunity[3]. However, some vaccines need to be stored at cryogenic temperatures that may not be deployable in developing areas of the world. Moreover, some mutations in SARS-CoV-2 genome may impact the effectiveness of vaccines to control the virus[4,5]. These underline the requirement for the parallel development of therapeutic options for SARS-CoV-2 treatment.

SARS-CoV-2 is an enveloped, positive-sense, single-stranded RNA virus with a large genome of about 30,000 nucleotides. The whole genome of SARS-CoV-2 is 96% identical to a bat coronavirus and closely related to SARS-CoV-1 with 80% sequence identity[6]. Two overlapping polyproteins, pp1a and pp1ab, are encoded by the replicase gene (ORF 1a/1b) that constitutes two-thirds of the genome. The remainder of the genome encodes for accessory and structural proteins, such as the spike glycoprotein, envelope protein, matrix protein and the nucleocapsid phosphoprotein[7]. pp1a and pp1ab are proteolytically digested into 15 non-structural proteins (NSPs) by the two viral proteases. The 33.8 kDa main protease (M$^{pro}$) or NSP5 is responsible for cleaving polyproteins at 11 cleavage sites giving NSP4-9 and NSP12-15. The released NSPs form the viral RNA polymerase complex are involved in replication and transcription of fresh virus in the host. Due to vital function in SARS-CoV-2 life cycle and absence of homologous proteins in human, M$^{pro}$ has been extensively explored by high-throughput screening of re-purposed druggable compounds[8] and fragments[9] to devise effective inhibitors aimed at arresting the growth of SARS-CoV-2 in host's cell.

SARS-CoV-2 M$^{pro}$ is a homodimeric enzyme consisting of three domains[8]. The substrate-binding site with a catalytic dyad of His41 and Cys145 is located between chymotrypsin-like domains I and picornavirus 3C protease-like domain II. Domain III plays an important role in M$^{pro}$ dimerization through a salt-bridge interaction between protomers. Several inhibitors and fragments have been co-crystallised and identified to block catalytic cavity[8–10].

Ebselen is an organoselenium molecule that can function as a glutathione peroxidase and peroxiredoxin mimic[11]. It has been shown to form a seleno sulphide bond with thiol groups of cysteine (Cys) on a number of proteins which results in anti-inflammatory, anti-microbial and neuroprotective effects[12–14]. Moreover, ebselen is being investigated in clinical trials as a potential therapy for stroke, hearing loss and bipolar disorder with good safety profiles with no adverse effects[15–17]. Recently, ebselen was identified in high-throughput screen as a potential hit of SARS-CoV-2 M$^{pro}$ inhibitor with an IC$_{50}$ between 0.67 and 2.1 μM[8,18]. Molecular dynamics simulations suggested that ebselen is able to bind at two probable sites[19]. One is at Cys145 within the catalytic cavity through a seleno sulphide bond, and another is at the dimerization region. However, no experimental data for the site of its binding in SARS-CoV-2 M$^{pro}$ has become available.

In our previous work, we have designed CNS penetrant ebselen-based derivatives and demonstrated their good neuroprotective effects and low cytotoxicity in cell-based and mouse models of motor neuron disease[20]. Here, ebselen and five derivatives were assessed for their inhibition of SARS-CoV-2 M$^{pro}$ and anti-coronaviral activity. Two of these ebselen-based selenium compounds exhibit greater inhibitory effectiveness than ebselen against M$^{pro}$ enzyme and SARS-CoV-2 replication. We show from co-crystallographic studies of M$^{pro}$ enzyme with ebselen and another potent compound (MR6-31-2) that these compounds solely bind at the M$^{pro}$ catalytic site by donating a selenium atom, forming a covalent bond and blocking the histidine-Cys catalytic dyad. We propose that the ebselen-enzyme drug protein adduct is hydrolysed by the conserved water in the catalytic pocket. The release of phenol by-product has been confirmed by mass spectrometry studies of M$^{pro}$ incubated with compounds. This intriguing selenation mechanism of inhibition and direct observation of covalent binding of the selenium atom together with sub-micromolar antiviral activity provides a rational for utilising ebselen as potential therapy and improving selenium-based compounds using the ebselen scaffold for greater anti-coronaviral activity.

## Results

**M$^{pro}$ enzymatic and antiviral activities of ebselen and derivatives.** In our previous study, ebselen and some selenium-based derivatives have were developed as neuroprotective agents in relation to motor neuron disease[20]. The co-crystalised structures of those compounds with superoxide dismutase 1 were proven to form a selenyl sulphide bond with Cys111 at dimer interface. Thus, ebselen and derivatives were considered for their reactivity with Cys145 and their potential for impairing proteolytic activity of M$^{pro}$ in an attempt to arrest the growth of SARS-CoV-2. A fluorescence resonance energy transfer assay was conducted to evaluate the inhibition level against M$^{pro}$ enzyme of ebselen and five other selenium-based derivatives. Figure 1 shows chemical structures, inhibitory curves against M$^{pro}$ and the half-maximal inhibitory concentrations (IC$_{50}$s) for each of the compounds. The data clearly demonstrates that these compounds including ebselen are potent M$^{pro}$ inhibitor with sub-micromolar levels of IC$_{50}$. Some compounds are twice as effective for M$^{pro}$ inhibition than the parent ebselen, especially MR6-7-2 and MR6-18-2. All of these compounds were also assessed for in vitro antiviral activity against SARS-CoV-2 infected primate Vero cells. All of the compounds tested were somewhat superior to ebselen with MR6-31-2 being nearly three times more effective with an EC$_{50}$ of 1.8 μM (Fig. 1g, Supplementary Fig. 1 and Supplementary Table 1). These results indicate clear on-target interaction of these compounds with M$^{pro}$ with significant inhibitory power for SARS-CoV-2 and as such potential for development as treatments for COVID-19 patients.

**Structures of M$^{pro}$ with ebselen and MR6-31-2 reveal selenium atom bound in catalytic site.** The interaction of ebselen and its derivative with M$^{pro}$ was directly visualised by co-crystallisation of organoselenium compounds with M$^{pro}$. The structures of ligand-free and M$^{pro}$ complexes with ebselen and MR6-31-2 have been solved at the resolution of 1.6–2.0 Å. The statistics of data collection and structure refinement is summarised in Table 1. All M$^{pro}$ structures have the same packing in C2 space group with only one M$^{pro}$ protomer found in asymmetric unit. The global structures of untreated and compound-treated M$^{pro}$ are almost identical with the root-mean-square deviations between 0.17 and 0.20 Å (Fig. 2a). The M$^{pro}$ catalytic site including Cys-histidine dyad of individual structures are given in Fig. 2b–d. Electron

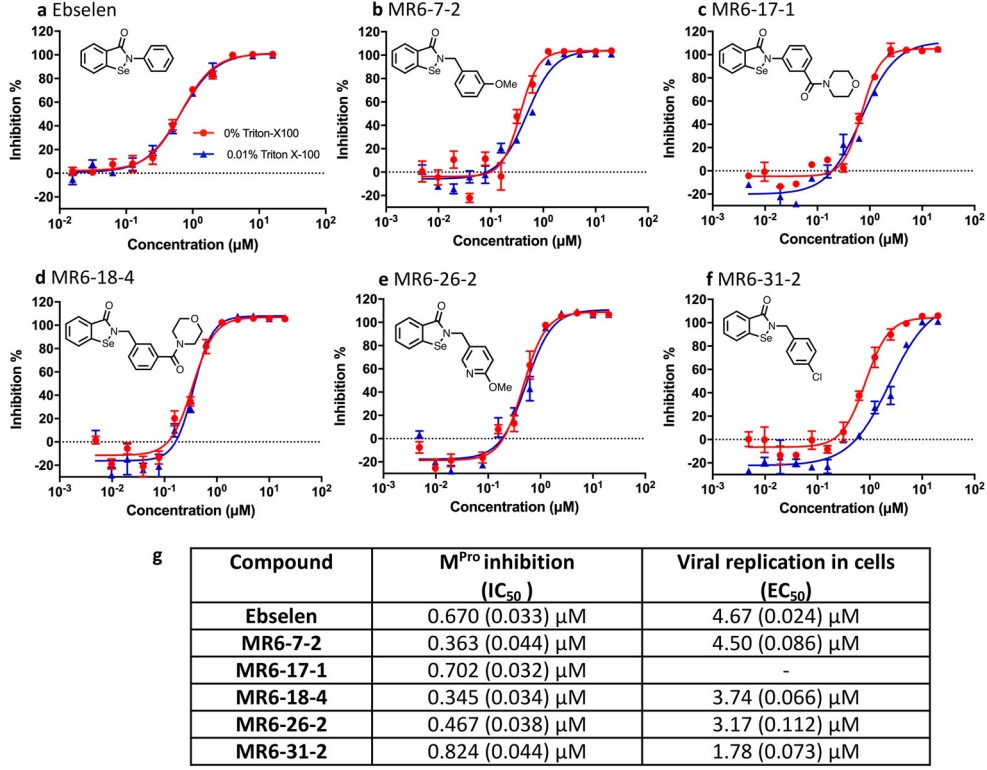

**Fig. 1 Chemical structures, in vitro M$^{pro}$ inhibition and cell-based antiviral assays of ebselen and five derivatives.** In vitro M$^{pro}$ inhibitory curves of **a** ebselen, **b** MR6-7-2, **c** MR6-17-1, **d** MR6-18-4, **e** MR6-26-2 and **f** MR6-31-2. Inhibition percentage plots are means of $n = 3$ measurements obtained over three independent experiments and error bars representing the standard error of the mean. **g** IC$_{50}$s of M$^{pro}$ inhibition and EC$_{50}$s of viral replication in Vero E6 cells. IC$_{50}$s and EC$_{50}$s are means (standard error of log(concentration)).

| Compound | M$^{pro}$ inhibition (IC$_{50}$) | Viral replication in cells (EC$_{50}$) |
|---|---|---|
| Ebselen | 0.670 (0.033) μM | 4.67 (0.024) μM |
| MR6-7-2 | 0.363 (0.044) μM | 4.50 (0.086) μM |
| MR6-17-1 | 0.702 (0.032) μM | - |
| MR6-18-4 | 0.345 (0.034) μM | 3.74 (0.066) μM |
| MR6-26-2 | 0.467 (0.038) μM | 3.17 (0.112) μM |
| MR6-31-2 | 0.824 (0.044) μM | 1.78 (0.073) μM |

**Table 1 Crystallographic data collection and refinement statistics of M$^{pro}$ and the complexes with ebselen and MR6-31-2.**

| | M$^{pro}$ | M$^{pro}$-ebselen | M$^{pro}$-MR6-31-2 |
|---|---|---|---|
| Data collection | | | |
| Space group | C2 | C2 | C2 |
| Cell dimensions | | | |
| $a$, $b$, $c$ (Å) | 115.35, 53.61, 44.98 | 113.20, 53.78, 44.92 | 114.00, 53.62, 44.54 |
| α, β, γ (°) | 90.00, 101.37, 90.00 | 90.00, 101.44, 90.00 | 90.00, 100.89, 90.00 |
| Resolution (Å) | 48.49-1.65 (1.68-1.65) | 48.39-2.05 (2.11-2.05) | 33.87-1.85 (1.89-1.85) |
| $R_{merge}$(%) | 3.7 (29.5) | 3.8 (58.5) | 7.5 (107.9) |
| $R_{pim}$(%) | 3.5 (26.7) | 3.6 (53.3) | 6.9 (97.1) |
| CC$_{1/2}$ | 0.999 (0.899) | 0.998 (0.699) | 0.997 (0.446) |
| $I$ /σ$I$ | 13.2 (2.2) | 12.4 (1.8) | 8.6 (1.6) |
| Completeness (%) | 99.9 (99.9) | 99.5 (100.0) | 99.9 (99.9) |
| Redundancy | 3.3 (3.4) | 3.4 (3.5) | 3.3 (3.2) |
| Refinement | | | |
| Resolution (Å) | 48.49-1.65 | 48.39-2.05 | 33.87-1.85 |
| No. reflections | 32,461 | 16,661 | 21,465 |
| $R_{work}$ /$R_{free}$(%) | 17.60/18.70 | 20.10/22.05 | 22.39/24.90 |
| No. atoms | | | |
| Protein | 2360 | 2324 | 2335 |
| Se$^{a}$ | 0 | 1 | 1 |
| Water | 250 | 88 | 139 |
| $B$-factors (Å$^2$) | | | |
| Protein | 15.58 | 50.12 | 36.18 |
| Se | – | 52.68 | 38.72 |
| Water | 37.88 | 51.39 | 39.47 |
| R.m.s. deviations | | | |
| Bond lengths (Å) | 0.0104 | 0.0066 | 0.0068 |
| Bond angles (°) | 1.712 | 1.489 | 1.572 |
| Ramachandran plot | | | |
| Preferred (%) | 97.37 | 97.35 | 96.69 |
| Allowed (%) | 2.30 | 2.32 | 3.31 |
| Outliers (%) | 0.33 | 0.33 | 0 |
| PDB code | 7BAJ | 7BAK | 7BAL |

$^a$Selenium occupancy is 0.60 (ebselen) and 0.80 (MR6-31-2).

density is clearly visible, allowing amino acid residues and water molecules to be defined accurately. Interestingly, a clear patch of electron density is observed between Cys145 and His41 in co-crystallised crystals of M$^{pro}$-ebselen and M$^{pro}$-MR6-31-2. This density is too strong to be a water, but the size is too small for the corresponding complete inhibitors. To identify the origin of this clear density, anomalous electron density map of selenium was calculated from diffraction data using X-ray at the wavelength near selenium absorption edge (0.97 Å). No selenium anomalous density is observed in ligand-free enzyme (Fig. 2b), but a strong anomalous density at 3σ is present only at the Cys-histidine catalytic dyad in ligand-treated M$^{pro}$ structures (Fig. 2c, d). Thus, selenium atom was modelled into the density with the distances of 2.2 Å away from Cys145 and His41. Calculated B-factor suggest that the occupancy of selenium in M$^{pro}$-ebselen and M$^{pro}$-MR6-31-2 crystals are 60% and 80%, respectively. These data are consistent with ebselen and MR6-31-2 primarily binding at M$^{pro}$ catalytic pocket and form a selenyl sulphide bond with Cys145. There was no density associated with the organic backbone of either ebselen or MR6-31-2 in the co-crystallographic structures. The position of the selenium is very similar to that obtained by molecular docking (Supplementary Fig. 2). As only selenium atom is observed in the enzyme's active site, it is the inactivation of Cys by selenium that results in the inhibition of M$^{pro}$ activity and viral replication. Interestingly, selenium binding does not affect the conformation of surrounding amino acid residues within the active site. Moreover, a conserved water molecule which is 3.6–4.0 Å away from Nε of His41, forming a hydrogen bond with the main chain of His164 was observed in all structures (Fig. 2b–d). The distance between His41 and the conserved water gets closer in M$^{pro}$-ebselen and M$^{pro}$-MR6-31-2 structures that is consistent with its role in hydrolysis of protein-compound adduct.

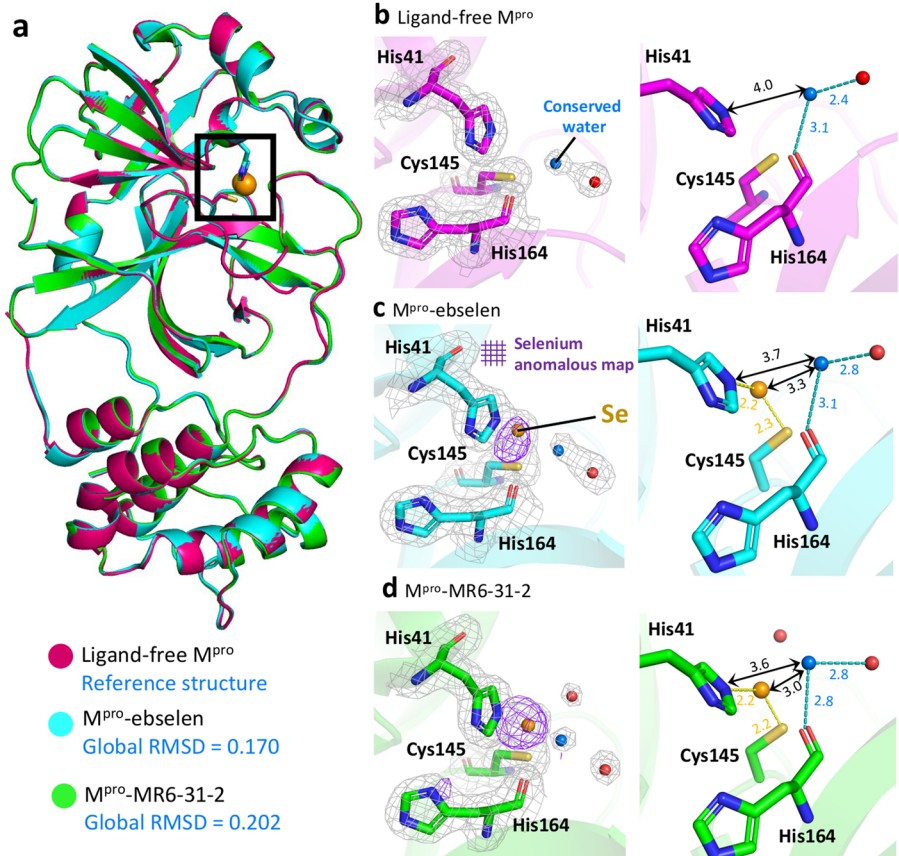

**Fig. 2 Crystallographic structures of ligand-free M^pro and the complexes with ebselen and MR6-31-2. a** Cartoon representation of superimposed structures of ligand-free M^pro (magenta), M^pro-ebselen (cyan) and M^pro-MR6-31-2 (green). The M^pro catalytic site is highlighted in a black box and the global root-mean-square derivations (RMSDs) of the compound-treated structures to ligand-free M^pro are given in blue texts. Close-up views of catalytic site of **b** ligand-free M^pro, **c** M^pro-ebselen and **d** M^pro-MR6-31-2. Electron density (2F$_o$–F$_c$) map is shown as grey mesh at 1σ. Anomalous signal of selenium is shown as purple mesh at 3σ. Selenium atom, conserved water and other waters are shown as orange, blue and red spheres, respectively. The close contacts below 2.5 Å and hydrogen bonds are shown as yellow and light blue dashes, respectively. The distances are illustrated by black double-headed arrows.

According to crystallographic evidence, we noted that ebselen and selenium-based derivatives have unusual mode of action by selenation of M^pro catalytic dyad.

**LC-MS characterization of salicylanilide by product generated by hydrolysis of ebselen.** Co-crystalised structures of M^pro with ebselen and MR6-31-2 demonstrated that the compounds inhibits M^pro by the selenation at Cys145 of catalytic dyad. This evidence suggests that the compounds may be hydrolysed within M^pro active site that releases its phenolic by-product (salicylanilide for ebselen, Fig. 3a). In order to identify the hydrolysis product derived from M^pro-ebselen adduct, LC/MRM-MS method was optimized using standard salicylanilide. A representative chromatogram from standard salicylanilide (85.2 ng/mL) is shown in Fig. 3a. Samples obtained from the incubation of ebselen showed strong peaks at 6.37 min corresponding to salicylanilide (Fig. 3b). The MS/MS spectrum of molecular ion at m/z 214 showed product ions at m/z 121 and m/z 94, which are attributed to the ions derived from the cleavage of the amide bond (Fig. 3c). To measure the levels of salicylanilide formed in the incubation, an 8-point calibration line was generated for salicylanilide in bovine serum albumin (BSA). The measured concentration of salicylanilide in these samples after 240 min was 1.28 ng/ml. The formation of salicylanilide in the incubation of M^pro with ebselen is time-dependent (Fig. 3d). LC-MS/MS analysis of the tryptic digest of ebselen-treated human glutathione S-transferase-pi

revealed a Cys-ebselen adduct with a mass addition of 274.996 amu, while no such mass addition was obtained with Cys145 in M^pro peptide FTIKGSFLNGSCGSVGF (Supplementary Fig. 3).

**Discussion**
From a chemical mechanism of action perspective, we fully expected to see the SARS-CoV-2 M^pro drug-adduct **2** from ebselen **1**, through nucleophilic attack of the Cys145 thiolate on the electrophilic selenium centre as shown in Fig. 4. Unlike other M^pro covalent inhibitors[8,21,22], the organic framework of ebselen was not present in the co-crystallographic structures with evidence of extrusion of selenium atom from the ebselen core at a Cys protease active site. We propose that His41 can assist a water-mediated attack on intermediate adduct **2** in an S$_N$Ar type hydrolysis reaction with intermediate **3** possibly stabilised in a manner akin to peptide hydrolysis tetrahedral intermediates within the oxyanion hole of the active site. With increased activity in the drug-design field in the covalent modification of catalytic and non-catalytic thiols, there have been several reports of aromatic warheads tuned with leaving groups (halides for example) to enable nucleophilic aromatic substitution, so the S$_N$Ar aspect of the proposed mechanism is with precedent[23–25]. Based on this mechanism, we would expect to see the generation of the hydrolysis product **4**. Using liquid chromatography mass spectrometry (LCMS) analysis of the SARS-CoV-2 M^pro and comparison with a commercial of **4**, we were able to show that **4** is

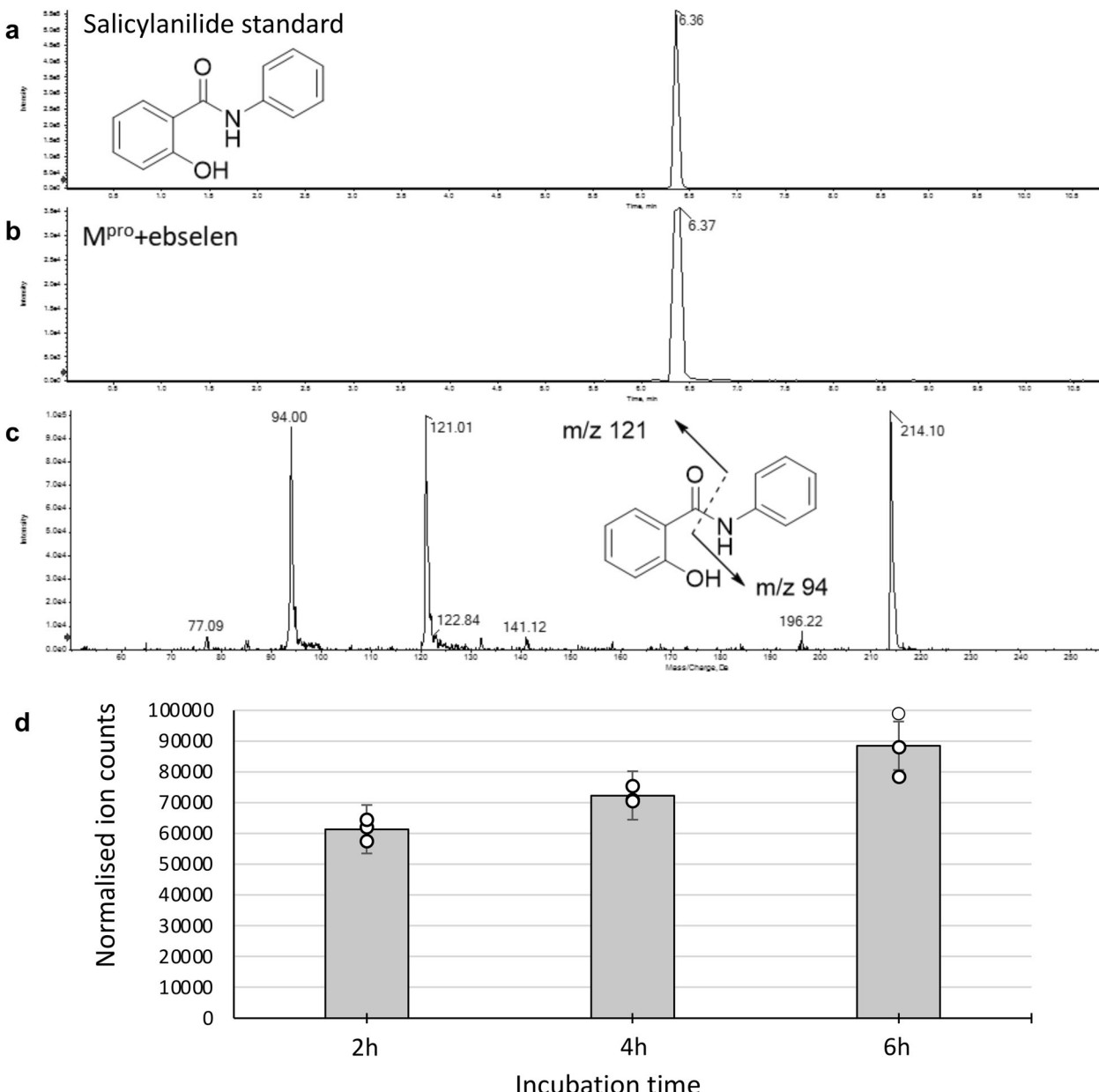

**Fig. 3 Representative chromatograph for salicylanilide standard and its formation in the incubation of M^pro with ebselen. a** The peak at 6.36 min retention time corresponds to salicylanilide standard (85.2 ng/mL). **b** Salicylanilide was detected in the incubation of M^pro with ebselen. **c** MS/MS spectrum shows the characteristic fragments derived from salicylanilide. **d** Time-course of salicylanilide formation. Bar chart represents means of $n = 3$ measurements obtained over three independent experiments and error bars representing the standard error of the mean.

**Fig. 4 Chemical mechanism for selenation of M^pro cysteine 145 by ebselen.** His41 assists a water-mediated attack on intermediate **2** leading to a hydrolysis reaction akin to peptide hydrolysis leading to the generation of the hydrolysis product **4**.

generated from ebselen by the enzyme in a time-dependent manner. This provides strong support for our proposed mechanism for selenation of the SARS-CoV-2 M$^{pro}$ active site.

We have succeeded in obtaining co-crystallographic structure of M$^{pro}$ grown with ebselen and its derivative MR6-31-2, showing selenium coordinates directly to Cys145 upon hydrolysis of the organoselenium compounds. The clear target engagement paves the way for the further development for more effective delivery to the catalytic Cys and greater inhibition whilst having an acceptable safety profile. The selectivity for Cys145 and sub-micromolar potency for M$^{pro}$ makes a strong case for benzoisoselenazolone to be integrated into known M$^{pro}$ inhibitor scaffolds. Though our study is of clear immediate interest for SARS-CoV-2, it has wider therapeutic applications of organoselenium compounds by chemical mechanism of the selenation of Cys of proteases in other current zoonotic *beta* coronaviruses and those that may emerge in the future.

## Methods

**Synthesis of compounds**. Ebselen was obtained from a commercial supplier (Sigma-Aldrich). Other lead compounds were produced and purified at Department of Chemistry, University of Liverpool. Details of the synthesis of ebselen-based derivatives have been described in Supplementary methods.

**Recombinant SARS-CoV-2 M$^{pro}$ production**. SARS-CoV-2 M$^{pro}$ gene (GenBank: MN908947.3, residues 3258-3569) containing modified human rhinovirus 3C protease (HRV-3C) cleavage site and 6xHis tag (SGVTFQGPHHHHHH) at C-terminal was cloned into pGEX-6P-1 vector at BamHI and XhoI sites using gene synthesis and cloning services (GenScript, USA). The plasmid was transformed into *E. Coli* strain BL21(DE3) and cultured at 37 °C in 2xYT broth until optical density at 600 nm reaches 0.8. M$^{pro}$ expression was induced by the addition of 0.5 mM isopropyl ß-d-1-thiogalactopyranoside followed by the incubation at 37 °C for 5 h. The bacteria pellet was harvested by centrifugation at $5000 \times g$, 4 °C for 20 min and then re-suspended in lysis buffer (20 mM Tris pH 7.8, 150 mM NaCl) before sonicated on ice. Cell lysate was collected by centrifugation at $30,000 \times g$, 4 °C for 30 min and then loaded onto a 5 mL NiNTA affinity column (HisTrap HP, GE Healthcare) pre-equilibrated with the lysis buffer. M$^{pro}$ bound to NiNTA resin was washed with 100 mL of 5 mM imidazole in lysis buffer and then eluted with a linear gradient of imidazole from 5 to 500 mM in lysis buffer, 100 mL. The fractions containing M$^{pro}$ were pooled together, mixed with recombinant His-tag HRV-3C, and dialysed against 20 mM Tris pH 7.8, 150 mM NaCl, 1 mM DTT at 4 °C overnight. The mixture containing M$^{pro}$ was re-loaded through fresh NiNTA resin to remove uncleaved protein and HRV-3C. The His-tag cleaved M$^{pro}$ in the flow-through was buffer-exchanged to 20 mM Tris pH 8 using Amicon Ultra centrifugal filter (MWCO. 10 kDa, Merck) and then loaded onto 5 mL Q Sepharose column (HiTrap Q HP, GE Healthcare). The column was eluted with 100 mL of a linear gradient from 0 to 200 mM NaCl in 20 mM Tris pH 8. The fractions containing pure M$^{pro}$ were buffer-exchanged to 20 mM Tris pH 7.8, 150 mM NaCl for activity assay and crystallisation or 25 mM ammonium bicarbonate pH 7.5 for LC-MS analysis. The concentration of M$^{pro}$ was determined by ultraviolent absorption at 280 nm using a molar extinction coefficient of 32,890 $M^{-1}cm^{-1}$.

**M$^{pro}$ inhibition activity assay**. The inhibition activity assays were performed using 0.2 μM M$^{pro}$, 20 μM substrate and serial-diluted tested inhibitors in 60 μL reaction buffer consisting of 50 mM Tris pH 7.3, 1 mM EDTA. Firstly, M$^{pro}$ was incubated with testing inhibitors at 30 °C for 15 min in reaction buffer. The reaction was then initiated by the addition of a FRET-based peptide substrate Mca–AVLQ↓SGFR-K(Dnp)K (GL Biochem), using wavelengths of 320 and 405 nm for excitation and emission, respectively. Fluorescence intensity was monitored with an EnVision multimode plate reader (PerkinElmer). Initial rate was obtained using the data from the first 10 min by linear regression. To exclude inhibitors possibly acting as aggregators, a detergent-based control was performed by adding 0.01% freshly made-up Triton X-100 to the reaction at the same time. The IC$_{50}$ was calculated by plotting the inhibition rate against various concentrations of testing inhibitor by using a four parameters dose–response curve in GraphPad Prism 8 software. All experiments were performed in triplicate.

**Antiviral activity assay**. A clinical isolate of SARS-CoV-2 (nCoV-2019BetaCoV/Wuhan/WIV04/2019) was propagated in Vero E6 cells, and viral titer was determined by 50% tissue culture infective dose (TCID$_{50}$) using immunofluorescence assay. Briefly, Vero E6 cells were fixed with 4% paraformaldehyde and permeabilised with 0.5% Triton X-100 before blocked with 5% BSA for 2 h at 25 °C. The blocked cells were incubated with the primary antibody of polyclonal antibody against viral nucleocapsid protein of a bat SARS-CoV[26] and followed by the second antibody of Alexa 488-labeled goat anti-rabbit (Abcam). The nuclei were stained with Hoechst 33258 dye (Beyotime) before visualised by fluorescence microscopy. For the antiviral assay, pre-seeded Vero E6 cells ($5 \times 10^4$ cells/well) were pre-treated with the different concentration of compound for 1 h and the virus was subsequently added (MOI of 0.01) to allow infection for 1 h. At 24-h post infection, the cell supernatant was collected and extracted viral RNA using MiniBEST Viral RNA/DNA Extraction Kit (Tanaka, #RR047A). Reverse transcription was conducted using PrimeScript RT Reagent Kit with gDNA eraser (Tanaka, #RR047A) to prepare cDNA template. qRT-PCR analysis was carried out on StepOne Plus Real-time PCR (Applied Biosystem) with TB Green Premix Ex Taq II (Tanaka, #RR820A). Receptor binding domain (RBD) of spike gene was amplified by PCR from the cDNA template with primers: RBD-F: 5′-GCTCCATGGCCTAATATTA CAAACTTGTGCC3′; RBD-R: 5′-TGCTCTAGACTCAAGTGTCTGTGGATCAC-3′, cloned into pMT/BiP/V5-His vector (Invitrogen) and used as the standard plasmid. A standard curve was generated by the determination of copy numbers from serial dilutions ($10^3$–$10^9$ copies) of the standard plasmid. The primers used for quantitative PCR were RBD-qF1: 5′-CAATGGTTTAACAGGCACAGG-3′ and RBD-qR1: 5′-CTCAAGTGTCTGTGGATCACG-3′[26]. PCR amplification was performed as follows: 95 °C for 5 min followed by 40 cycles consisting of 95 °C for 15 s, 54 °C for 15 s, 72 °C for 30 s. For cytotoxicity assays, pre-seeded Vero E6 cells were treated with appropriate concentrations of compound. After 24 h, the relative numbers of surviving cells were measured by the CCK8 (Beyotime) assay in accordance with the manufacturer's instructions. All experiments were performed in triplicate, and all the infection experiments were performed at biosafety level-3.

**Crystallisation and structure determination**. Ebselen and other compounds were prepared as 250 mM stocks in DMSO. 0.1 mM purified M$^{pro}$ was incubated with 1 mM compound at 4 °C overnight before concentrated to 5–15 mg/mL protein. Hanging crystallisation drops were set by mixing of 3 μL of M$^{pro}$, 2.4 μL of reservoir solution (200 mM ammonium chloride, 5% glycerol and 16–20% poly-ethylene glycol (PEG) molecular weight 3350) and 0.6 μL of 1/2560 diluted micro-seed stock. Micro-seed stock was prepared by crushing M$^{pro}$ crystals obtained from an initial hit (well A9 of JCSG+ screen: 200 mM NH$_4$Cl, 20% PEG3350, Molecular Dimensions) with glass seed beads (Hampton Research). The crystallisation drops were placed against 300 μL corresponding reservoir at 19 °C allowing vapour diffusion. Plate crystals of M$^{pro}$ appeared among precipitation within a week. The crystals were cryo-protected in 25% glycerol in reservoir solution before snap-frozen in liquid nitrogen. X-ray diffraction experiments were carried out at 100 K using 0.9795 Å beam on I04 beamline of Diamond Light Source, UK. Identification of selenium in active site was made by anomalous x-ray diffraction measurement in the same crystal using 0.9795 Å wavelength at 3.0 Å resolution. The data were integrated by using Xia2 DIALS[27] and scaled by using Aimless in CCP4 suite[28]. Phase problem was solved by molecular replacement with MOLREP[29] in CCP4 suite using a SARS-CoV-2 M$^{pro}$ structure (PDB: 6Y2E) as an initial model. Structure models were edited manually in COOT[30] and refined by using Refmac5[31] in CCP4 suite. Geometry and quality of final models were validated by using MolProbity[32]. All molecular structures were visualised using Pymol software.

**Liquid chromatography mass spectrometry (LCMS)**. 10 mg/mL M$^{pro}$ in 25 mM ammonium bicarbonate pH 7.5 (20 μL) was incubated with 1 mM ebselen at 37 °C for 0, 2, and 4 h. At the end of incubation, 2.5 mM acetaminophen (10 μL) was added as an internal standard to normalize extraction. Then, loading and compounds of interest were extracted by adding ice-cold acetone (250 μL). Standard curve was constructed by spiking salicylanilide (concentration range: 0.1–0.8 μM) into 10 mg/mL BSA solution. After centrifugation at $16,100 \times g$ for 20 min, the extracts were transferred to clean tubes and evaporated in a Speed Vac and reconstituted in 50 μL 30% ACN/0.1% formic acid. A total of 10 μL of samples and standards were analysed immediately by a QTRAP 5500 mass spectrometer (AB Sciex) coupled with an Ultimate 3000 HPLC system (Dionex, ThermoScientific) and a Kinetex C18 column (2.6 μM, C18, 50 mm × 2.1 mm, Phenomenex). The MS experiments were conducted using electrospray ionization with positive ion detection. A gradient programme of acetonitrile (5% for 1 min; 5–95% over 5 min; 95% for 2 min; 95–5% over 0.1 min; 5% for 4 min) in 0.1% formic acid (v/v) was applied at a flow rate of 300 μL/min. The multiple reaction monitoring transitions for each analyte were as following: salicylanilide 214.2/121.1 and 214.2/95; acet-aminophen, 152.1/108.1; other MS parameters, such as voltage potential and collision energy were optimized to achieve great sensitivity. Data acquisition and quantification were performed using Analyst 1.5 software, Multi-Quant 3.0 (AB Sciex) and Microsoft Excel.

**Reporting summary**. Further information on research design is available in the Nature Research Reporting Summary linked to this article.

## Data availability

Source data are provided with this paper. Other data are available from the corresponding authors upon reasonable request. Crystal structures of ligand-free M$^{pro}$ and M$^{pro}$ with ebselen and MR6-31-4 have been deposited in the Protein Data Bank under accession codes: 7BAJ, 7BAK and 7BAL, respectively.

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

## Acknowledgements

This work was supported by the BBSRC's.IAA award 168064. Some of the developments arose from the ALS Association's grant (WA1128). We acknowledge the University's silver command team and the institute's technical team for making arrangements so that experimental works could be undertaken despite COVID-19 restrictions. We would like to acknowledge the support of the staffs and management of the Diamond Light Source (Didcot, UK) for the beamtimes and operations at the facility (proposal: mx27113).

## Author contributions

S.S.H. and P.M.O. conceived the work. K.A. purified protein and performed crystallographic experiments. X.M. performed mass spectrometry experiment. M.R. synthesised ebselen-based compounds. W.S. and L.Z. performed and analyzed the antiviral assay in biosafety level 3. Z.J. performed the enzymatic inhibition assay. Z.J.L. participated in crystallographic data analysis. W.S., Z.J., Y.Z., Z.R., Z.L, H.Y. and L.Z. analyzed and discussed the antiviral and inhibition data. All authors read and approved the final version of the manuscript.

## Competing interests

The authors declare no competing interests.
