## [Peer Review File · Nature Communications]

REVIEWER COMMENTS

Reviewer #1 (Remarks to the Author):

In response to the SARS-CoV-2 pandemic, Ampornnanai et al investigated inhibition of the Main protease (Mpro) as a potential antiviral. The authors mention that ebselen was identified as a potential inhibitor of SARS-CoV-2 Mpro through a high throughput screen. Because the authors have previously designed ebselen-based derivatives against a motor neuron disease, it was logical to test the derivatives against SARS-CoV-2 Mpro. The authors report IC50s as low as 345 nM and EC50s as low as 1.78 μ M. In addition to the inhibition data, the authors provide novel structures of SARS-CoV-2 Mpro with the selenium ion bound to the catalytic cysteine, validated by anomalous scattering. The authors also purified the salicylanilide byproduct and validated that through mass spectrometry. This work demonstrates that the selenium atom is attached through a hydrolysis reaction. This is different than their previous work where ebselen was shown to covalently bind to proteins like SOD1 through a selenosulfide bond. Overall, this work was well conceived and this data would be valuable to the scientific community. What this story lacks is an explanation of the SAR data. Can the authors explain how the different inhibitor modifications affect potency, based on interactions with relevant active site residues? Cocrystallization with an inactive variant (like C145A) would provide a definitive answer, but simply docking the inhibitors in silico would be useful. Additional questions and comments are given below.

Questions/Comments

-Abstract: Despite the word limit, for quick reference I suggest highlighting your biochemical data in the abstract. Consider briefly mentioning your derivatives achieve sub-micromolar IC50 and single-digit-micromolar EC50.

-Line 68-70. Regarding the fact that ebselen was identified through a high throughput screen, the author should report the observed potency from that screen. A range of reported values may be more informative, as ebselen potency has been reported several times, including that ebselen dramatically loses potency in the presence of DTT (DOI: 10.1021/acsptsci.0c00130).

-Line 89 + Line 95: In the author's previous work, ebselen and 14 ebselen-derivatives were tested, but in this study only includes ebselen and 5 of those compounds. Why wasn't the whole panel tested, even if just testing IC50s? If that data exists, it should be included.

-Figure 1a-f: The authors provide the IC50 curves, but the EC50 curves should also be included.

-Figure 1g: The authors must report standard error on the IC50 and EC50 values.

-Line 111-112: I agree the global alignment is almost identical, but this could be quantified by RMSD.

-Figure 2c-d. It's unclear why the H2O is included when it's 5-5.5 Å away? How far away is the other conserved water? Also, the yellow dashes need to be defined in the figure legend; are they hydrogen bonds, are they distances, both? Lastly, is one of these conserved waters

proposed to participate in hydrolysis, as described in Fig 4?

-Figure 4. Is it known if substitutions to the benzoisoselenazolone impact this mechanism, and thereby potency?

-Discussion: Regarding wider therapeutic applications, are the authors suggesting the benzoisoselenazolone should be integrated into known Mpro inhibitor scaffolds?

Reviewer #2 (Remarks to the Author):

An organoselenium drug Ebselen and its derivatives are broadly reported as inhibitors of viral, bacterial and human enzymes. It has been commonly believed that these inhibitors act by covalent modification of a Cys residue, typically that which is present in the active site. The ring opening forms an Ebselen-Cys adduct via S-Se linkage. In the manuscript NCOMMS-20-51331A-Z, submitted by Ampornnanai et al. to Nature Communications, an unprecedented mechanism of Ebselen reaction with SARS-CoV-2 Mpro has been suggested. The authors proposed histidine-mediated hydrolysis of the preliminary formed S-Se covalent complex. As the result of the intermediate decomposition, enzyme-Cys-SeH and salicylanilide are produced. Clear evidences were provided to support the hypothesis: the crystal structures for enzyme selenation, and chromatography analyses of the extracts after incubation for the anilide release.

The work is definitely worth of publication in Nature Communications. It presents a fully novel and non-typical reactivity mechanism. Although described for SARS-CoV-2 Mpro, the emerging target for the coronavirus treatment, the data have a broad impact on inhibition studies on other proteases/hydrolases. SARS-CoV-2 PLpro (Weglarz-Tomczak et al. Scientific Reports 3640, 11, 2021) or bacterial urease (Macegoniuk et al. J. Med. Chem. 8125, 59, 2016) can be representative examples.

As the suggested mechanism of nucleophilic substitution in the aromatic system can be considered as quite controversial I wonder if the authors tried its further validation by a non-enzymatic approach. Reacting of a benzoisoselenazolone with a cysteine derivative (or another thiol) under appropriate acid-base catalysis should allow to perform this in glass.

Reviewer #3 (Remarks to the Author):

In their contribution, Hasnain and colleagues elucidate the inhibitory mechanism of ebselen, which was previously found to inhibit SARS-CoV-2 at submicromolar concentrations. The authors present three protein structures, one unliganded and two complex structures, obtained by co-crystallization of ebselen and the above derivative with SARS-CoV-2. In the complexes, electron density could not be detected for the covalently bound ligands, but only for parts of the ligands. Supported by anomalous X-ray diffraction experiments, the authors finally assigned the corresponding electron density to a covalently bound selenium atom. The authors provide a reasonable reaction mechanism: a nucleophilic attack on the active site thiolate gives rise to a seleno-enzyme intermediate. In a second reaction step, this is cleaved by the attack of a water molecule, which in principle follows a nucleophilic substitution mechanism, with the 2-hydroxy-N-phenylbenzamide being the leaving group. The authors show by MS/MS experiments that this intermediate is formed. The manuscript is mostly well written, but one can see that different passages have been written by different people. In addition, not all corrections have been fully adopted, such as in

line 56: of/in and in line 89: have/were. In addition, salicylanilide is misspelled several times.

Before accepting for publication, the authors should comment on the following:

The SARS-CoV-2 inhibitory activity of ebselen has already been published by some of the authors themselves, and in a more recent publication Weglarz-Tomczak et al. reported the activity of ebselen derivatives against SARS-CoV-2 (Scientific Reports 2021, similar but not the same derivatives). In their original publication, the authors stated that ebselen binds covalently to SARS-CoV-2 Mpro, as evidenced by a mass shift of 275. Could the authors comment on this, at first sight, contradictory result?

The authors state that the formation of 2-hydroxy-N-phenylbenzamide is time-dependent (Fig. 3d, with the first time point recorded after 2 hours). In the enzymatic assay, the compounds are pre-incubated for 15 minutes, and overnight for the determination of the crystal structure. The question thus arises whether the hydrolysis of the seleno-enzyme complex, which has a mass difference of 275 to the native protein, is due to the prolonged reaction time compared to the IC50 determination. In this context, it would be helpful to also investigate the postulated two-step mechanism of inhibition by enzyme kinetics.

Reviewer #1 (Remarks to the Author):

In response to the SARS-CoV-2 pandemic, Amporndanai et al investigated inhibition of the Main protease (Mpro) as a potential antiviral. The authors mention that ebselen was identified as a potential inhibitor of SARS-CoV-2 Mpro through a high throughput screen. Because the authors have previously designed ebselen-based derivatives against a motor neuron disease, it was logical to test the derivatives against SARS-CoV-2 Mpro. The authors report IC₅₀s as low as 345 nM and EC₅₀s as low as 1.78 μM. In addition to the inhibition data, the authors provide novel structures of SARS-CoV-2 Mpro with the selenium ion bound to the catalytic cysteine, validated by anomalous scattering. The authors also purified the salicylanilide byproduct and validated that through mass spectrometry. This work demonstrates that the selenium atom is attached through a hydrolysis reaction. This is different than their previous work where ebselen was shown to covalently bind to proteins like SOD1 through a selenosulfide bond. Overall, this work was well conceived and this data would be valuable to the scientific community. What this story lacks is an explanation of the SAR data. Can the authors explain how the different inhibitor modifications affect potency, based on interactions with relevant active site residues? Cocrystallization with an inactive variant (like C145A) would provide a definitive answer, but simply docking the inhibitors in silico would be useful.

Response: It is difficult to be certain about how inhibitor modifications affect potency as we only saw selenium atom between His41 and Cys145 in the catalytic site. We tentatively propose that potency is related to the more effective delivery of the molecule to the catalytic cysteine and have clarified this further. We have taken up reviewer's suggestion of molecular docking and show that the higher potency of compounds with the methylene linker (MR6-7-2, MR6-18-4 and MR6-26-2) than ebselen and MR6-17-1 may arise from the ability of methylene linker to establish additional interactions with nearby pocket. The chlorine substituent at the tail group of MR6-31-2 seems to reduce fitness score and whilst the more flexible methoxy and carbonyl morpholine substituents are more favourable in term of fitness scores correlating well with IC₅₀ (Table S1). Thus, the enhancement of inhibition level of an ebselen-based compound is likely achieved mainly through effectiveness of delivery of the compound, initial non-covalent stabilisation to the catalytic pocket preceding the nucleophilic attack of Cys 145. A larger data set will be required to see if one can correlate activity with scoring functions in the future but see initial finding from docking results as part of a response to comment below.

Additional questions and comments are given below.

Questions/Comments

-Abstract: Despite the word limit, for quick reference I suggest highlighting your biochemical data in the abstract. Consider briefly mentioning your derivatives achieve sub-micromolar IC₅₀ and single-digit-micromolar EC₅₀.

We have done

-Line 68-70. Regarding the fact that ebselen was identified through a high throughput screen, the author should report the observed potency from that screen. A range of reported values may be more informative, as ebselen potency has been reported several times, including that ebselen dramatically loses potency in the presence of DTT (DOI: 10.1021/acsptsci.0c00130).

Response: Ebselen is characterized as an enzyme mimic, carrying out the function of the selenoenzyme, GSH peroxidase. The reviewer may be concerned about its function in the reducing environment of the cytosol. First, we noticed that the authors of the quoted paper carried out their experiments under extremely reducing conditions (4 mM DTT) in the paper (DOI: 10.1021/acsptsci.0c00130), which does not reflect the physiological environment in the

cells. Second there are some research groups who have found that ebselen can work *in vivo* independent of its function mimicking GSH peroxidase (Ref. 1-2). The Bogoyo group reported that ebselen could target the cysteine protease domain (CPD) within the *Clostridium difficile* major virulence factor toxin B (TcdB) for treating this particular bacterial infection. The cysteine at the active site of the CPD can be covalently modified by ebselen, which resulted in inhibition of the release of the toxic glucosyltransferase domain (GTD). They also demonstrated that ebselen treatment reduced the pathology of *Clostridium difficile* infection in mice via oral gavage (Ref. 1). Though there is an argument whether ebselen targets the GTD as well or not, this study supports the idea that modification of cysteine protease may work *in vivo*. Another example is by Valente *et al.* who showed that ebselen could inhibit early viral post-entry events of the HIV-1 life cycle by targeting HIV-1 capsid. In their paper, ebselen covalently binds the HIV-1 capsid CTD, likely via a selenylsulfide linkage with Cys198 and Cys218 (Ref. 2). Our situation is similar to the first case since ebselen can modify the C145 of SARS-CoV-2 M^{pro}. In our case, the plaque reduction assay (published in Nature) and RT-qPCR testing in this study did confirm that ebselen and its derivatives could inhibit viral replication in the cells. Nonetheless, we cannot entirely exclude the possibility that ebselen could act on other binding sites in M^{pro} (Ref. 3) as additional benefits.

1. Bender, K. O. *et al.* A small-molecule antivirulence agent for treating *Clostridium difficile* infection. *Science translational medicine* **7**, 306ra148-306ra148 (2015).

2. Thenin-Houssier, S. *et al.* Ebselen, a Small-Molecule Capsid Inhibitor of HIV-1 Replication. *Antimicrobial Agents and Chemotherapy* **60**, 2195-2208, doi:10.1128/aac.02574-15 (2016).

3. doi: 10.1126/sciadv.abd0345

We have added reported range of values IC₅₀ in our revised version.

-Line 89 + Line 95: In the author's previous work, ebselen and 14 ebselen-derivatives were tested, but in this study only includes ebselen and 5 of those compounds. Why wasn't the whole panel tested, even if just testing IC₅₀s? If that data exists, it should be included.

Response: Five compounds tested in this work are the best performers in term of SOD1 cys111 binding and stabilisation according to our previously publication. Thus, the whole panel was not tested for M^{pro} and viral inhibition.

Compound	ΔT_m (A4V SOD1)	IC ₅₀ (M ^{pro})
Ebselen	+8.77°C	670 nM
MR6-7-2	+7.77°C	363 nM
MR6-17-1	+9.90°C	702 nM
MR6-18-4	+8.67°C	345 nM
MR6-26-2	+8.08°C	467nM
MR6-31-2	+6.36°C	824 nM

-Figure 1a-f: The authors provide the IC₅₀ curves, but the EC₅₀ curves should also be included.

We have included EC₅₀ curves in FigureS1 in the supplementary information.

-Figure 1g: The authors must report standard error on the IC₅₀ and EC₅₀ values.

IC₅₀ values were calculated by using Prism 8 which reports standard error in log (concentration). We have added standard errors to our Figure 1g.

-Line 111-112: I agree the global alignment is almost identical, but this could be quantified by RMSD.

Response: Global RMSDs of ligand-treated Mpro structures compared to ligand-free Mpro have been added to Figure 2.

-Figure 2c-d. It's unclear why the H₂O is included when it's 5-5.5 Å away? How far away is the other conserved water? Also, the yellow dashes need to be defined in the figure legend; are they hydrogen bonds, are they distances, both? Lastly, is one of these conserved waters proposed to participate in hydrolysis, as described in Fig 4?

Response: There is a conserved water molecule that is 3.6-4.0Å away from N ϵ of His41 and makes H-bond with main chain of His164 and another water molecule. We have revised the figure and marked the conserved water. We have used dashed lines in different colours and double-headed to present interactions and distances, respectively.

-Figure 4. Is it known if substitutions to the benzoisoselenazolone impact this mechanism, and thereby potency?

Response: We have referenced molecular docking as reported in our Ebiomedicine paper (Figure S7) and have included additional molecular docking results of the wild-type Mpro and C145A in-silico mutant in the supplementary information, clearly showing the anchoring role of cysteine for Se.

-Discussion: Regarding wider therapeutic applications, are the authors suggesting the benzoisoselenazolone should be integrated into known Mpro inhibitor scaffolds?

Response: Yes, as its selectivity for cys145 and potency for Mpro is clear. As a reactive fragment, potency could well be enhanced further by the attachment of additional non-covalent binding peptidomimetic fragments to an ebselen like core- with a range of covalent small molecules being evaluated such possibilities could be investigated moving forward.

Reviewer #2 (Remarks to the Author):

An organoselenium drug Ebselen and its derivatives are broadly reported as inhibitors of viral, bacterial and human enzymes. It has been commonly believed that these inhibitors act by covalent modification of a Cys residue, typically that which is present in the active site. The ring opening forms an Ebselen-Cys adduct via S-Se linkage. In the manuscript NCOMMS-20-51331A-Z, submitted by Amporndanai et al. to Nature Communications, an unprecedented mechanism of Ebselen reaction with SARS-CoV-2 Mpro has been suggested. The authors proposed histidine-mediated hydrolysis of the preliminary formed S-Se covalent complex. As the result of the intermediate decomposition, enzyme-Cys-SeH and salicylanilide are produced. Clear evidences were provided to support the hypothesis: the crystal structures for enzyme selenation, and chromatography analyses of the extracts after incubation for the anilide release. The work is definitely worth of publication in Nature Communications. It presents a fully novel and non-typical reactivity mechanism. Although described for SARS-CoV-2 Mpro, the emerging target for the coronavirus treatment, the data have a broad impact on inhibition studies on other proteases/hydrolases. SARS-CoV-2 PLpro (Weglaz-Tomczak et al. Scientific Reports 3640, 11, 2021) or bacterial urease (Macegoniuk et al. J. Med. Chem. 8125, 59, 2016) can be representative examples. As the suggested mechanism of nucleophilic substitution in the aromatic system can be considered as quite controversial. I wonder if the authors tried its further validation by a non-enzymatic approach. Reacting of a

benzisoselenazolone with a cysteine derivative (or another thiol) under appropriate acid-base catalysis should allow to perform this in glass.

Response: We have done the MS experiment of ebselen reaction in other protein (human glutathione S-transferase, GSTP) containing free cysteines. This result indicated that protein-ebselen adduct was formed but no hydrolysis occurred due to lack of histidine in catalytic dyad unlike in Mpro (Fig.S2A). Thus, we can conclude that His41 in Mpro may be essential for this unprecedented mechanism as proposed in Fig 4. Biomimetic *in vitro* replication of the conditions within the active site of a cysteine protease active site and the “super reactivity” of the catalytic thiol is virtually impossible to achieve with biologically relevant thiols such as glutathione and N-acetyl cysteine. For example, the unique acid/ base catalysis can see up to 20,000 x difference in reactivity between glutathione and a site directed electrophile (see for example work performed with alpha halo carbonyl inhibitors (in studies with the enzymes papain and cathepsin B, see eg. Eur J Med Chem (1992) 27,865-873). Even if such reactions were to work after extensive reaction trialling, the results would be open to question as relevant. From a chemical reactivity perspective, we will examine the possibility in due course as part of on-going programme.

Reviewer #3 (Remarks to the Author):

In their contribution, Hasnain and colleagues elucidate the inhibitory mechanism of ebselen, which was previously found to inhibit SARS-CoV-2 at submicromolar concentrations. The authors present three protein structures, one unliganded and two complex structures, obtained by co-crystallization of ebselen and the above derivative with SARS-CoV-2. In the complexes, electron density could not be detected for the covalently bound ligands, but only for parts of the ligands. Supported by anomalous X-ray diffraction experiments, the authors finally assigned the corresponding electron density to a covalently bound selenium atom. The authors provide a reasonable reaction mechanism: a nucleophilic attack on the active site thiolate gives rise to a seleno-enzyme intermediate. In a second reaction step, this is cleaved by the attack of a water molecule, which in principle follows a nucleophilic substitution mechanism, with the 2-hydroxy-N-phenylbenzamide being the leaving group. The authors show by MS/MS experiments that this intermediate is formed.

The manuscript is mostly well written, but one can see that different passages have been written by different people. In addition, not all corrections have been fully adopted, such as in line 56: of/in and in line 89: have/were. In addition, salicylanilide is misspelled several times.

Response: We have edited and checked “salicylanilide” throughout the revised version.

Before accepting for publication, the authors should comment on the following: The SARS-CoV-2 inhibitory activity of ebselen has already been published by some of the authors themselves, and in a more recent publication Weglarz-Tomczak et al. reported the activity of ebselen derivatives against SARS-CoV-2 (Scientific Reports 2021, similar but not the same derivatives). In their original publication, the authors stated that ebselen binds covalently to SARS-CoV-2 Mpro, as evidenced by a mass shift of 275. Could the authors comment on this, at first sight, contradictory result?

Response: We found the MS data of ebselen binding to Mpro varied greatly with experimental and instrument conditions. In this study, ebselen-Cys145 adduct with a mass shift of 275 was not detected, which is consistent with the observation in the structural study. However, this adduct (M+245) was detected on Cys47 in human glutathione S-transferase pi (GSTP) and Cys16 in Mpro, indicating an adduct is stable without the presence of neighbouring catalytic histidine residues. We therefore searched for selenium-Cys145 adduct which will have a mass shift of 79.99 but was unable to confirm this adduct by MS data. We speculated the peptide containing selenium-Cys145 adduct could potentially form a cross-linked peptide with another

peptide containing a cysteine residue, which is not identified by searching software. We therefore include the following in the revised manuscript:

The binding of ebselen to Mpro appeared to be variable. Ebselene was previously reported to covalently bind to Cys145 in Mpro peptide FTIKGSFLNGSCGSVGF with a mass addition of 275 amu². However, this adduct was not detected on Cys145 in this study, which is consistent with the finding in the structural study. The proposed hydrolysis reaction could result in an adduct with a mass addition of 79.99 amu where selenium is attached to Cys145. The peptide containing this adduct could potentially form a cross-linked peptide with another cysteine containing peptide, which is generally difficult to be identified by searching software. Interestingly, ebselen was found to covalently bind to Cys16 with a mass addition of 274.996 amu, indicating this adduct is stable without the presence of neighbouring catalytic histidine residues nearby (Figure S3).

There have been a number of publications showing *in silico* ebselen binding to cys145 or other pockets. However, none of them gives clear experimental evidence to validate covalent binding of ebselen to Mpro such as mass spectrometry or co-crystallised structure. The publication by Weglarz-Tomczak et al. reported ebselen inhibitory activity against SARS-CoV-2 papain-like protease (PLpro). Although ebselen could form covalent bond with cys111 in PLpro with the mass shift evidence, the active sites of PLpro and Mpro are totally different

The authors state that the formation of 2-hydroxy-N-phenylbenzamide is time-dependent (Fig. 3d, with the first time point recorded after 2 hours). In the enzymatic assay, the compounds are pre-incubated for 15 minutes, and overnight for the determination of the crystal structure.

Response: Unfortunately, the nature of crystallisation experiment is such that crystallisation is unpredictable. Reviewer may have misunderstood as crystallisation can take from days to weeks. Even in the cases reported here, diffraction quality crystals are obtained after several days and in some cases even longer.

REVIEWER COMMENTS

Reviewer #1 (Remarks to the Author):

The authors sufficiently addressed my comments, questions, and suggestions from the initial review. The integration of the feedback from the three reviewers made this manuscript more robust and clarified several points. I could quibble that the docking and SAR discussion could belong in the main text, but if a reader were interested in that analysis they will seek it out in the supplementary information. Overall, I think this work will be of great interest to those designing inhibitors against SARS-CoV-2 Mpro and other cysteine proteases.

Reviewer #2 (Remarks to the Author):

The authors of the revised version of manuscript NCOMMS-20-51331B appropriately addressed all the comments and remarks of the referees which were made to the original version of the paper. New data have been provided and thoroughly discussed in the responses. Thus, I sustain my previous judgement of a high scientific value of the contribution, and recommend its publication in Nature Communications.